# Pitch Gradation by Ion-Dragging Effect in Polymer-Stabilized Cholesteric Liquid Crystal Reflector Device

**DOI:** 10.3390/polym12010096

**Published:** 2020-01-04

**Authors:** Xiaowen Hu, Weijie Zeng, Xinmin Zhang, Kai Wang, Xiaoling Liao, Xinshuai Jiang, Xiao-Fang Jiang, Mingliang Jin, Lingling Shui, Guofu Zhou

**Affiliations:** 1SCNU-TUE Joint Lab of Device Integrated Responsive Materials (DIRM), National Center for International Research on Green Optoelectronics, South China Normal University, Guangzhou 510006, China; xwhu@m.scnu.edu.cn (X.H.); sftianhai@163.com (W.Z.); zxm681126@163.com (X.Z.); m15815403849@163.com (X.L.); guofu.zhou@m.scnu.edu.cn (G.Z.); 2Guangdong Provincial Key Laboratory of Optical Information Materials and Technology & Institute of Electronic Paper Displays, South China Academy of Advanced Optoelectronics, South China Normal University, Guangzhou 510006, China; jxs0315@163.com (X.J.); jinml@scnu.edu.cn (M.J.); shuill@m.scnu.edu.cn (L.S.); 3Department of Materials Science and Engineering, Pennsylvania State University, University Park, PA 16802, USA; kqw5449@psu.edu; 4Shenzhen Guohua Optoelectronics Tech. Co. Ltd., Shenzhen 518110, China

**Keywords:** infrared reflector, cholesteric liquid crystal, reflection band broadening, polymer photocuring, direct current (DC) bias

## Abstract

An IR reflector based on polymer-stabilized cholesteric liquid crystal (PSCLC) can selectively tune IR light reflection for smart window application. Broadening the reflection bandwidth to block more IR heat radiation requires the expansion of the pitch distribution in the PSCLC. Traditional attempts using ex situ direct current (DC) bias upon an already polymerized PSCLC reflector usually require a sustaining potential difference holding the pitch gradient of the reflector. Removing the DC bias will lead to a reflect bandwidth comeback. Here, we have developed an in situ DC curing strategy to realize an irreversible reflect bandwidth broadening. Briefly, a DC bias was used to drive the redistribution of impurity cations, which can be captured by the ester group of oligomers, during the photopolymerization. During the slow polymerization process, such trapped cations will drag the oligomers towards the cathode and compress the pitch length near the cathode before the oligomers form the long polymer chain. Consequently, a frozen pitch gradient by such an in-situ-electric-field-assisted dynamic ion-dragging effect leads to the formation of a pitch gradient along the electrical field direction. After removing the DC bias, the as-cured polymer is observed to have frozen such a gradient pitch feature without recoverable change. As a result, the PSCLC reflector exhibits steady bandwidth broadening of 480 nm in the IR region, which provides the potential for saving energy as a smart window.

## 1. Introduction

Today, global overheating has triggered various threats to our planet and modern civilization [1]. Since 1970, Earth’s heat content has risen at a rate of 6 × 10^21^ joules a year [2], the equivalent of the energy output of about 190,000 nuclear power plants. The sun is the primary source of Earth’s heat and according to the National Aeronautics and Space Administration (NASA) about 51% of solar radiation is absorbed by the land and ocean, while 30% is reflected back to space [2]. In solar spectra, heat mainly comes from the light in the IR region in the wavelength range 700 to 2500 nm, accounting for 50% of solar energy [3]. Reflecting IR radiation will cool down an object and protect it from overheating. Cholesteric liquid crystal (CLC) exhibits a spontaneous helical structure and is capable of reflecting at most 50% of unpolarized natural light, making it a good candidate for an IR heat reflector. Such a CLC design is omnipresent in natural living creatures with evolved collagen, cornea, chitin, or cellulose in order to selectively reflect photon radiation and survive in a harsh environment. For example, the crab *Carcinus maenas* has developed a cholesteric-structured cuticle to sufficiently reflect IR radiation and avoid overheating, as the crab spends a lot of time on beaches [4,5]. In the artificial world, CLC materials for IR heat reflectors have also been well-conceptualized and encouraged by the increasing demand for buildings that contain a comfortable indoor temperature. Smart windows that can selectively allow visible light to pass through but reflect IR heat exhibit huge application potential not only with regard to saving energy for domestic cooling but also for compromising global warming issues.

Organic CLC materials have attracted a broadening amount of attention as they are advantageous in tuning the transmittance of IR radiation depending on environmental conditions [6]. In addition, compared to commercial inorganic IR reflectors, non-metallic organic CLC will not block or interfere with wireless communication signals such as radio frequency (RF), cyber, or cellular signals, etc. [7]. However, to sufficiently reflect IR heat, a wide reflection bandwidth is required. The broader the reflection band, the more IR heat will be reflected. The bandwidth (∆λ) of the reflected light is defined by $∆\lambda =\Delta n\cdot P=\left(n_{e}-n_{0}\right)\cdot P$, with $\Delta n=\left(n_{e}-n_{0}\right)$, $n_{e}$, $n_{0}$, and P being the birefringence, extraordinary refractive index, ordinary refractive index, and the helical pitch of the CLC, respectively [8,9]. Normally, since the Δn of colorless organic materials is usually less than 0.3, the band width of a single-pitch CLC in the IR region is limited to a few tens of nanometers [10], making it insufficient for heat reflection. In this regard, broadening the reflection bandwidth requires a non-uniform pitch distribution or a pitch gradient in the CLC gel films. Prior strategies for multiplication of pitch in CLC IR reflectors have mainly relied on a functional dopant that is sensitive to light to induce a nonuniform degree of polymerization along the vertical direction (the direction perpendicular to the film plane). For example, Broer et al. introduced a high UV extinction dopant to induce a gradient of UV radiation and hence obtained gradient UV curing throughout the thickness direction [11,12]. Seiji et al. [13] and Chen et al. [14] utilized sugar derivative and chiral photoisomers, respectively to induce the trans–cis photoisomerization that can produce a reversible change in the helical pitch of CLC. Additional strategies including low-dose UV initiated curing [15], thermal controlled polymerization [16], reactive interfacial layer coating [17], mechanical strain [18], magnetic field [19], and ex situ electric field [20,21,22,23,24] assisted pitch change have also been reported. In particular, applying an external electric field on a polymer-stabilized cholesteric liquid crystal (PSCLC) reflector cell can modify the orientation of the CLC molecules and also drive the translational motion of the polymer network, inducing a pitch change in the cell. However, such a pitch change is highly reversible and the broadened pitch will recover as soon as the holding bias is removed, as the polymer network has already been cured before the direct current (DC) electric field is applied. This is counter to the motivation of a wide bandwidth for IR heat reflector implementation.

In this study, we introduce an in-situ-DC-electric-field-assisted polymerization that takes advantage of the in-situ-electric-field-assisted cation-dragging effect on oligomers during the slow polymerization process. The dynamic cation-dragging effect further compresses the pitch length near the cathode, leading to a pitch gradient along the electrical field direction within the PSCLC. After removing the DC bias, the as-cured polymer network and pitch gradient are still observed to be very well maintained. Consequently, the PSCLC reflector which is based on a polymer-stabilized CLC displays a steady bandwidth broadening of 480 nm in the IR region.

## 2. Materials and Methods

### 2.1. Materials

The LC mixture consisted of both reactive (polymerizable) and non-reactive (non-polymerizable) mesogens mixed with a photo-initiator. The nematic non-reactive LC mixture MLC-2079 with negative dielectric anisotropy was chosen to keep the LC director oriented in a planar direction upon application of a DC electric field. The CLC mixture was prepared by mixing together 78.7% wt % MLC-2079 (Δε = −6.7, Merck, Darmstadt, Germany), 9 wt % diacrylate monomer RM82 (Merck, Darmstadt, Germany), 11.3 wt % chiral dopant S811 (Merck), and 1 wt % photo-initiator Irgacure-651 (Ciba Specialty Chemicals (China) Ltd, Shanghai, China). All materials were used as received without further purification. Figure 1 shows the chemical structures of RM82, S811, and Irgacur-651. For the investigation of different photo-initiator concentrations with regard to cell performance, we changed the concentration of Irgacure-651 from 0.1% to 0.3%, 0.5%, 1%, and 1.5%, respectively.

### 2.2. Device Preparation

Firstly, the cleaned indium tin oxide (ITO) coated glasses were irradiated by ozone for 20 min in an ozone tank (BZS250GF-TC, Huiwo, Shenzhen, China) to make them more hydrophilic. Then, the inner surfaces of the ITO glass substrates were treated with polyvinyl alcohol (PVA) alignment layers, which involved rubbing with a plush cloth in parallel directions. Two obtained ITO glasses with PVA layers were placed face to face and bonded by a mixture of spacers (SiO_2_) and UV glue with a weight ratio of 1:99. The gap between the two ITO glasses was 25 µm, as decided by the spacers. Under UV irradiation for 1 min, a liquid crystal cell was obtained. Meanwhile, differently sized spacers were also used (5, 15 and 40 µm) to produce cells with different cell gaps.

The LC mixture was stirred thoroughly at 60 °C to ensure uniformity before use. The mixture was filled into the cell by capillary force. The sample cells obtained above were subjected to an in-situ DC electric field during polymerization. Here, the UV light source used for polymerization was Model BZS250GF-TC from Shanghai Qunhong (Shanghai, China), whose UV light intensity was fixed at 13 mW/cm^2^ centered at a wavelength of 365 nm. The polymerization time was 15 min. The whole preparation process was conducted in the yellow-light area.

### 2.3. Characterization

Optical characterization of the sample cells was obtained using unpolarizing spectrophotometry (Lambda 950, PerkinElmer, Shanghai, China) in transmission mode at normal incidence. Optical microscope photographs of PSCLC sample cells were observed with a polarizing light microscope (POM, LEICA DM2700P, Leica, Solms, Germany) at room temperature. Impedance spectra were measured with an impedance analyzer (TONGHUI ELECTRONICS, TH2828, Tonghui, Changzhou, China).

## 3. Results and Discussion

Figure 2 schematically illustrates the application of ex situ and in situ electric fields on the polymerization of the PSCLC cell. Briefly, an as-prepared CLC reflector consisting of a CLC gel containing impurity ions, non-reactive LC, a reactive LC monomer, and a photoinitiator was sandwiched between two parallel ITO substrates coated with alignment layer PVA (Figure 2(ai)), which displayed a single pitch length feature. The impurity ions in the mixture might have originated from synthetic and/or purification steps (catalysts, salts, moisture, and dust) [25,26], alignment layers [27], and/or degradation of LC molecules [28]. By shining UV light to initiate the polymerization for the reactive LC monomer, a cured network was able to be achieved within the cell to stabilize the LC orientation directors or freeze the ordered structures (Figure 2(aii)). Normally, applying an external electric field on such an already-made CLC cell can adjust the pitch length in the PSCLC cell, and it is well-known that impurity ions trapped by an ester group in a polymer chain are prone to be driven to the corresponding electrodes and thus induce a translational motion of the polymer network [22,29,30,31].

Consequently, pitch length modulation (Figure 2(aiii)) with a highly recoverable feature (Figure 2(aiv)) was able to be obtained. In comparison, we applied an in situ DC bias during the slow polymerization process in the CLC cell. As shown in Figure 2(bi), in the initial state, the CLC cell displayed a single pitch feature. By simultaneously applying a DC bias and UV light-initiated slow polymerization, the oligomers which were formed first and which contained the ester group were able to capture the impurity cations in the CLC by Columbic interaction. The trapped cations anchored at the short oligomer chains were able to further drag them towards the cathode under the in situ DC bias (Figure 2(bii)). After completing the whole polymerization process, the short oligomer chains connected with each other to form a polymeric network. Meanwhile, the in-situ-electric-field-assisted dynamics of the charged oligomers make the CLC stacking much more compact near the cathode (Figure 2(biii)). In this way, a pitch gradient was realized using the cation-dragging effect strategy. Moreover, as the polymerization was completed under a DC bias, the density of the polymeric network had a gradient along the thickness direction which regulated the CLC stacking in a pitch gradation manner. Removing the DC bias induced neither the comeback of the network nor the pitch gradient (Figure 2(biv)).

As the dynamic of the ion-dragging effect under the in situ DC bias played a key role in forming the non-uniform polymeric network, the strength of the electric field was of great importance in inducing the pitch gradient. Figure 3a compares the transmission spectra between the PSCLC cell treated with in situ DC electric fields of different intensities from 0 (the control) to 3.2 V/µm. It is clear that a larger electric field strength led to a larger reflect bandwidth. For example, as the field strength increased from 0 to 1.6 to 3.2 V/µm, the band width increased from 126 to 161 to 278 nm, respectively. Moreover, such a reflect band is only located in the IR region after a wavelength of 800 nm, indicating no transparency loss in the visible region (and maintenance of over 90% transparency). We further quantified the IR reflecting ability by plotting the full-width at half-maximum (FWHM) of the reflection bandwidth (extracted from the transmittance groove) versus the electric field intensity. As shown in Figure 3b, increasing the electric field intensity monotonously enlarged the reflection bandwidth. This was due to the dragging effect of the trapped cations in the polymer network that stretched the short oligomer chains moving toward the cathode during polymerization. The motion and distortion of the polymer network would have compressed the helical pitch of the LC on the negative electrode side and simultaneously stretched the helical pitch on the positive side. After polymerization there was a pitch gradient throughout the thickness of the cell and thus an enhanced reflection bandwidth. Figure 3c shows POM images of the PSCLC samples treated by different DC biases. It can be seen clearly that all samples exhibited an appropriate Grandjean planar texture under the POM, suggesting that the in situ electric field treatment did not affect the original cholesteric phase or its orientation in the LC molecules.

To further verify the hypothetical mechanism, we carried out a series of experiments by changing different variables in the CLC mixture. Fundamentally, during the slow polymerization, the short-chain oligomers which were first produced were able to trap the impurity cations forming a charged segment within the oligomers. Such a charged component can be driven to the corresponding electrode under an in situ DC bias. In the limit case of there being no monomer or no polymerization process, no cation-dragging effect will be present even under an in situ electric field, and, hence, the pitch length stays invariant. In order to prove this, we fabricated a CLC cell without either monomers or photo-polymerization with different DC bias treatments. Figure 4a,b show the transmittance spectra of these CLC cells, whose reflection band did not change regardless of the different electric field intensities. Figure 4c,d quantify the FWHM of the bandwidth in dependence on the DC electric field intensity. It is clear that there was no difference in bandwidth. These results suggest that the presence of both monomers and polymerization is required for the cation-dragging effect.

In the typical case where the polymerization process and the trapped cation-dragged oligomer motion simultaneously exist within the system, a much more complicated dynamic process is present in the cell in which the reaction rate, drifting velocity, and cell thickness will synergistically affect the final morphological configuration. Fundamentally, the dynamic competition between the DC-assisted oligomer drifting velocity and the polymerization rate are the key to the proposed pitch gradient formation in the resultant PSCLC cell. Faster drifting or a slower polymerization rate will facilitate nonuniform polymerization or the network gradient, as there will be enough time for the oligomer to accumulate at one side of the electrode before further polymerization into longer chains occurs. Thus, we tuned both the polymerization rate and the drifting rate to investigate the underlying physics. The polymerization rate can be modulated by changing the initiator concentrations to adjust the propagation reaction rate [32]; the drifting rate can be adjusted by changing the DC bias intensity to tune the field acceleration coupled by its time integration [33]. Figure 5a displays the reflection bandwidth summarized as a function of in situ bias from 0 to 3.2 V/um with different initiator concentrations from 0.1% to 1.5%. It is clear that as the DC bias increased there was an overall increase in reflect bandwidth, regardless of the initiator concentration. This was due to the stronger electric field forces that made it easier for the oligomer to drift towards the cathode. On the other hand, at different DC biases the bandwidth had different dependence on the initiator concentrations. For example, at a lower bias of 0.8 V/μm, the bandwidth of the PSCLC cell using 0.5%, 1.0%, and 1.5% initiator concentrations displayed a similar value of 137 nm, while at a higher bias of 3.2 V/μm, the PSCLC cell using 0.5%, 1.0%, and 1.5% initiator concentrations exhibited a monotonous increased bandwidth of 256, 278, and 290 nm, respectively. Theoretically, the lower the initiator concentration, the slower the polymerization experienced, which will provide enough time for the oligomer to drift towards the front-side cathode and consequently for a denser polymer network and then a larger reflection bandwidth to occur. Moreover, in real cases the light penetration for initiating the polymerization should also be taken into consideration. In fact, prior reports have revealed the importance of UV light penetration on gradient pitch formation in CLC systems [34]. Here, the UV initiate light comes from the top cathode and the achiral diacrylate (RM82) molecule first undergoes polymerization near the top cathode. Due to the consumption for prior polymerization near the top cathode, RM82 diffuses from bottom to top. When the polymerization rate is much faster than the dynamic motion of RM82, RM82 is frozen into the network before moving to the top, and a less gradient-like network is formed. Hence, a lower initiator concentration gives rise to a broader bandwidth when the cell is without in situ DC bias treatment. To confirm this, we further investigated the reflection bandwidth upon different initiator concentrations (different polymerization rates) without an in situ electric field. As shown in Figure 5b, as the initiator concentration increased from 0.1% to 1.5%, the reflection bandwidth exhibited a monotonous decrease from 136 to 120 nm. This is in distinct contrast to that of the cell using in situ electric field treatment (Figure 5a), where since the concentration increased from 0.5 to 1.0 to 1.5 wt %, the bandwidth increased from 256 to 278 to 290 nm at the in situ DC bias of 3.2 V/μm, showing much larger bandwidths than in the case without in situ electric field treatment. It seems that with high in situ DC bias treatment, a lower polymerization rate due to the smaller initiator concentration does not dominate the bandwidth change. It is probably true that the impurity cation concentration, which could be enlarged by adding more initiator molecules, could further complicate the case. We further measured the impedance spectra of the cells with different photo-initiator concentrations (Figure 5c). We found that the CLC system with the highest initiator concentration of 1.5% showed the lowest impedance, suggesting the largest ion concentration. This could be beneficial for the broadening of the reflection band because more ions could be trapped at the oligomer and induce a more intensive dragging effect to evoke a more inhomogeneous network. Overall, it can be concluded that the competition relationship between the polymerization rate and the dynamic motion of the charged oligomer determines the final bandwidth performance.

The dynamic motion of charged oligomers under a vertical in situ electric field is limited by cell thickness. Previous studies have also revealed the importance of cell thickness in affecting the broadening effect [35]. We further investigated the dependence of the reflect bandwidth on cell thickness with an in situ DC bias treatment. The transmittance spectra of cells with different cell thickness which were treated with an increased in situ DC electric field are shown in Appendix A (Appendix A). Figure 5d summarized the bandwidth of PSCLC cells with different thicknesses from 5 to 40 μm under different in situ DC curing bias treatments. As the cell thickness increased, there was a wider reflection bandwidth. Since the bandwidth is proportional to the pitch gradient, a wider bandwidth suggests a larger pitch gradient of the PSCLC. At a larger thickness, the degree of the trapped cation dragging effect can reach a larger scale, as there is more distance for the cations to move. Thus, the dragging effect compresses and stretches the polymer at an extended level, rendering the resultant network more nonuniform. Notably, under an electric field of 3.2 V/µm and the largest cell thickness of 40 µm, a maximal reflection bandwidth of 480 nm was obtained.

So far we have demonstrated that an in situ DC electric field that can fix the polymerization in a gradient manner induces an irreversible bandwidth broadening. Based on this finding, we further applied an ex situ electric field to see if there was additional room to tune the bandwidth upon our already broadened samples. The PSCLC cell which was polymerized under an in situ electric field of 2.4 V/µm was used for further ex situ DC bias testing. Figure 6a shows the transmittance spectra of the PSCLC cell under different ex situ DC biases. It can be seen as the ex-situ DC bias increased, a bandwidth broadening was still exhibited. We also quantified the reflection bandwidth as a function of ex situ DC bias, as shown in Figure 6b, and found that the reflection bandwidth was even able to increase from 220 to 540 nm after applying an ex situ bias of 3.6 V/μm. Thus, our in-situ-treated PSCLC cell still showed itself to be capable of an additional tunability with an ex situ bias. This can be understood with regard to the additional movement of the already frozen network under the ex situ bias, which is consistent with reports of ex situ bias inducing reversible band broadening [22]. However, here the starting cell had already been broadened and the ex situ bias gave an extra extent for electrically tuning the bandwidth onto the next level. As a result, the bandwidth was able to be enlarged to 540 nm under an ex situ field of 3.6 V/μm.

## 4. Conclusions

In summary, we have reported in this work on the preparation of a PSCLC with a negative dielectric anisotropy in which the selective reflection bandwidth could be greatly broadened by an in situ curing DC electric field. Briefly, during the slow polymerization process, the oligomers which were first produced were able to trap the impurity cations and drift towards the cathode under such an in situ DC bias. Thus, the oligomer drifting and further polymerization into long chains occurred simultaneously, resulting in a final nonuniform network which regulated the pitch length of the CLC at a larger scale. Consequently, reflection bandwidth broadening was able to be achieved using this in-situ-bias-assisted trapped cation-dragging effect. This dynamic played a key role in the competition between the drifting and polymerization rate and determined the final device performance. In this study, using this strategy, we achieved a primitive reflection bandwidth increase to 480 nm (from 126 nm), proving the aforementioned concepts to be true. We believe that by properly tuning the in situ curing condition and the dynamics of drifting, the IR reflection regulation will further improve, which is of great importance for practical applications for green-cooling such as smart windows.

## Figures and Tables

**Figure 1 polymers-12-00096-f001:**
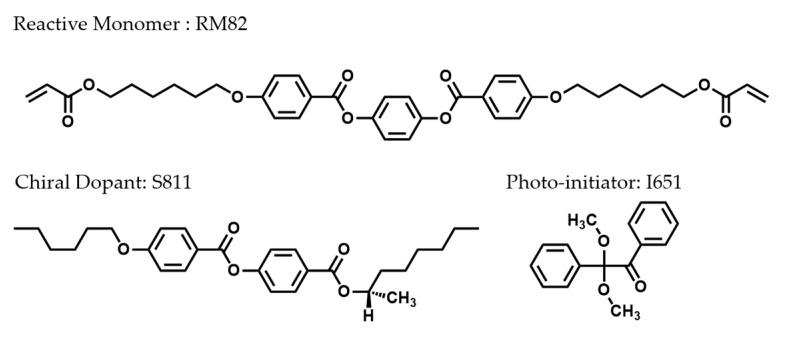
The molecular structures of RM82, S811, and Irgacure-651.

**Figure 2 polymers-12-00096-f002:**
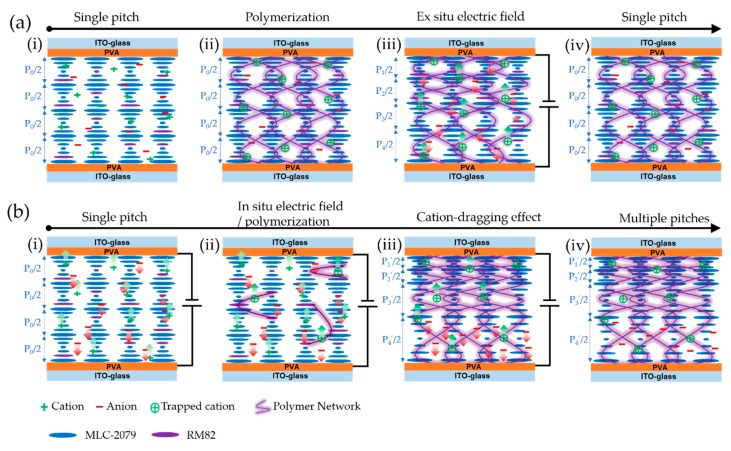
Schematic illustration of pitch length modulation in CLC cell under (**a**) ex situ electric field and (**b**) in situ electric field. Legend: ITO, indium tin oxide; PVA, polyvinyl alcohol.

**Figure 3 polymers-12-00096-f003:**
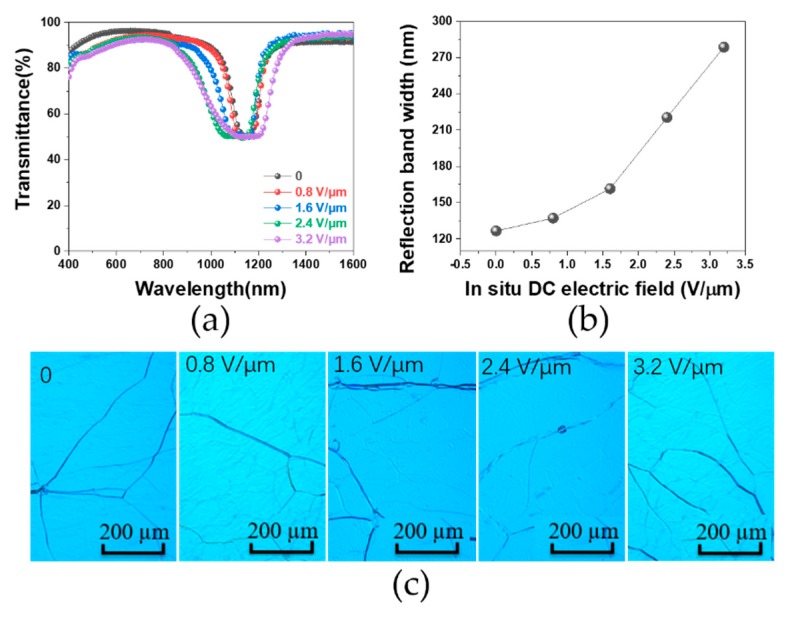
(**a**) Transmission spectra of polymer-stabilized cholesteric liquid crystal (PSCLC) cells treated with different in situ electric fields; (**b**) reflection bandwidth of the PSCLC cells versus in situ direct current (DC) electric field; (**c**) polarizing light microscope (POM) images of the PSCLC cells treated with different in situ electric fields.

**Figure 4 polymers-12-00096-f004:**
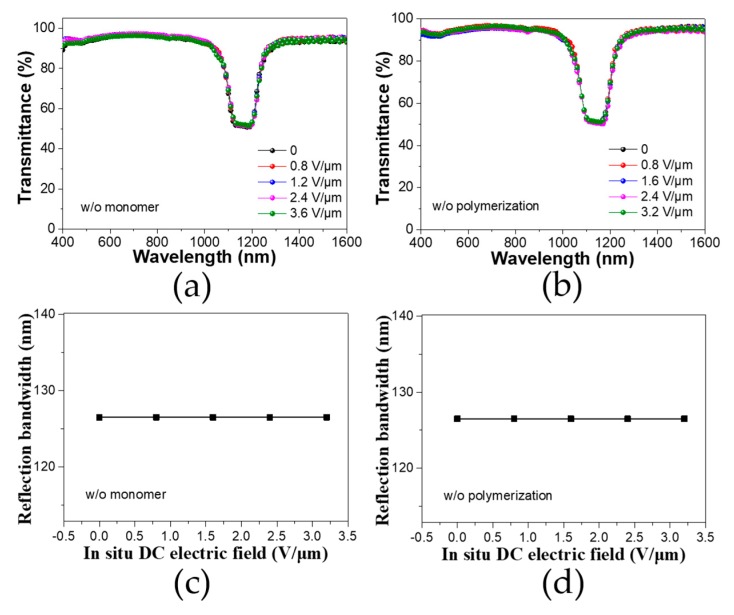
Transmittance spectra of a CLC cell without either monomers (**a**) or photo-polymerization (**b**) under different DC bias treatments; reflection bandwidth of the CLC cell without either monomers (**c**) or photo-polymerization (**d**) under different DC biases. Legend: *w*/*o*, without.

**Figure 5 polymers-12-00096-f005:**
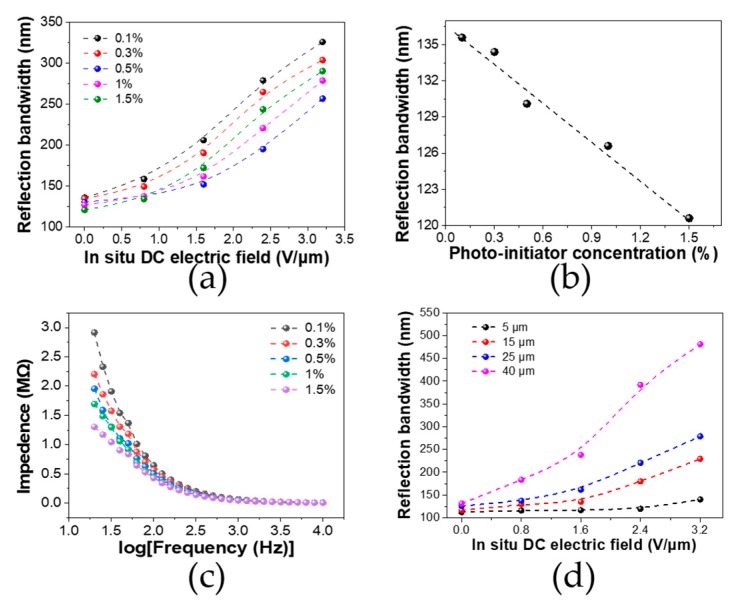
(**a**) Reflection bandwidth of the cell with different initiator concentrations treated with an in situ electric field; (**b**) reflection bandwidth of the cell without an in situ DC electric field as a function of different initiator concentrations; (**c**) impedance spectra of the cells with different photo-initiator concentrations; (**d**) reflection bandwidth of PSCLC cells with different thicknesses treated with an in situ DC electric field.

**Figure 6 polymers-12-00096-f006:**
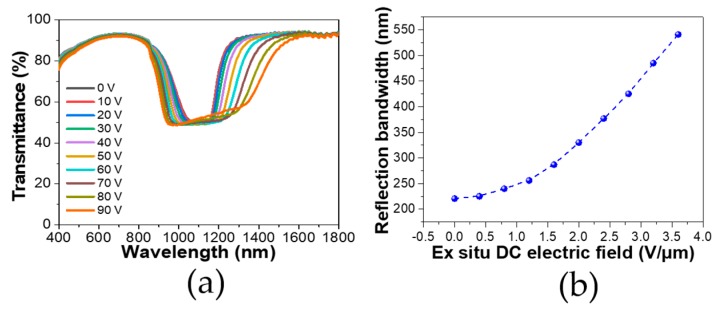
(**a**) Transmittance spectra of PSCLC cell under different ex situ DC electric fields. A PSCLC cell treated with an in situ DC electric field of 2.4 V/µm was chosen for the application of an ex situ DC electric field. (**b**) Reflection bandwidth of the cell under different ex situ DC electric fields.

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
