# Peer review of "Pitch Gradation by Ion-Dragging Effect in Polymer-Stabilized Cholesteric Liquid Crystal Reflector Device"

_polymers, 2020, doi:10.3390/polym12010096_

Round 1

Reviewer 1 Report

The authors demonstrated an in-situ DC electric field (EF) assisted polymerization which uses EF assisted cation-dragging effect on the oligomers during the polymerization process. The dynamic cation-dragging effect leads to a pitch gradient along the electrical field direction within the PSCLC. The obtained PSCLC reflector based on polymer stabilized CLC displays a steady bandwidth broadening of 480 nm in the IR region.
This manuscript can be accepted for publication to Polymers after minor revision.

1)

the literature review should be done more carefully. The phenomenon of ionic impurities and their interactions with electric field in liquid crystals are well studied problem, especially in the area of Liquid Crystals Devices (LCD). AC electric field were also used for acceleration of phase separation in liquid crystals/polymer blends, e.g. Ziebacz et al.:

https://onlinelibrary.wiley.com/doi/abs/10.1002/cphc.200900505

2) 

Can authors assess/calculate or measure the concentration of the ionic impurities in their samples?

3)

If not, maybe influence of the ionic impurities on the performance of the process can be verified with artificial doping of the samples with some organic or inorganic salts?

4)

I think that calculation the the electric field inside the cell would be a nice thing, if authors have access to COMSOL or any other FEM software it would be a nice addition.

Reviewer 2 Report

Comments attached.
